# Cheminformatics-Based Design and Synthesis of Hydroxyapatite/Collagen Nanocomposites for Biomedical Applications

**DOI:** 10.3390/polym16010085

**Published:** 2023-12-27

**Authors:** Mohamed Aaddouz, Khalil Azzaoui, Rachid Sabbahi, Moulay Hfid Youssoufi, Meryem Idrissi Yahyaoui, Abdeslam Asehraou, Mohamed El Miz, Belkheir Hammouti, Sergey Shityakov, Mohamed Siaj, Elmiloud Mejdoubi

**Affiliations:** 1Laboratory of Applied Chemistry and Environment, Team: Mineral Chemistry of Solids, Department of Chemistry, Faculty of Sciences, Mohammed 1st University, P.O. Box 717, Oujda 60000, Morocco; hafid.youssoufi@gmail.com (M.H.Y.); ee.mejdoubi@gmail.com (E.M.); 2Laboratory of Engineering, Electrochemistry, Modeling and Environment, Faculty of Sciences, Sidi Mohamed Ben Abdellah University, Fez 30000, Morocco; k.azzaoui@yahoo.com; 3Euromed Research Center, Euromed Polytechnic School, Euro-Mediterranean University of Fes, P.O. Box 15, Fes 30070, Morocco; hammoutib@gmail.com; 4Higher School of Technology, Ibn Zohr University, Quartier 25 Mars, P.O. Box 3007, Laayoune 70000, Morocco; 5Laboratory of Bioresources, Biotechnology, Ethnopharmacology and Health, Faculty of Sciences, Mohammed Premier University, Oujda 60000, Morocco; iy.meryem@ump.ac.ma (M.I.Y.); asehraou@yahoo.fr (A.A.); 6Laboratory of Molecular Chemistry, Materials and Environment (LCM2E), Multidisciplinary Faculty of Nador, University Mohamed I, Nador 60700, Morocco; elmiz.mohamed@ump.ac.ma; 7Department of Bioinformatics, Würzburg University, 97074 Würzburg, Germany; shityakoff@hotmail.com; 8Department of Chemistry, Université du Québec à Montréal, NanoQAM/QCAM, Montréal, QC H3C 3P8, Canada

**Keywords:** antimicrobial, bioavailability, biocomposite, collagen, co-precipitation, hydroxyapatite

## Abstract

This paper presents a novel cheminformatics approach for the design and synthesis of hydroxyapatite/collagen nanocomposites, which have potential biomedical applications in tissue engineering, drug delivery, and orthopedic and dental implants. The nanocomposites are synthesized by the co-precipitation method with different ratios of hydroxyapatite and collagen. Their mechanical, biological, and degradation properties are analyzed using various experimental and computational techniques. Attenuated total reflection–Fourier-transform infrared spectroscopy, thermogravimetric analysis, and X-ray diffraction unveil the low crystallinity and nanoscale particle size of hydroxyapatite (22.62 nm) and hydroxyapatite/collagen composites (14.81 nm). These findings are substantiated by scanning electron microscopy with energy-dispersive X-ray spectroscopy, confirming the Ca/P ratio between 1.65 and 1.53 and attesting to the formation of non-stoichiometric apatites in all samples, further validated by molecular simulation. The antimicrobial activity of the nanocomposites is evaluated in vitro against several bacterial and fungal strains, demonstrating their medical potential. Additionally, in silico analyses are performed to predict the absorption, distribution, metabolism, and excretion properties and the bioavailability of the collagen samples. This study paves the way for the development of novel biomaterials using chemoinformatics tools and methods, facilitating the optimization of design and synthesis parameters, as well as the prediction of biological outcomes. Future research directions should encompass the investigation of in vivo biocompatibility and bioactivity of the nanocomposites, while exploring further applications and functionalities of these innovative materials.

## 1. Introduction

Biocomposites are materials that combine biological and synthetic components to mimic the properties of natural biological materials, such as bones and teeth. These materials have a wide range of biomedical applications, such as tissue engineering, drug delivery, and orthopedic and dental implants [1,2]. Biocomposites have several advantages over conventional biomaterials, such as their ability to replicate the mechanical properties of natural biological materials, such as strength and toughness [3,4]. Moreover, biocomposites can be designed to be biocompatible, which means that they do not trigger an immune response or cause other adverse reactions in the body [5,6]. Furthermore, biocomposites can promote cell growth and differentiation, which can enhance the long-term success of the implant [7].

One of the most popular types of biocomposites is the chitosan-based composite, which is widely used in tissue engineering and wound healing [8,9,10]. However, HAp-based composites are extensively used in biomedical applications, such as bone repair and regeneration, because of their biocompatibility and osteoconductivity [11,12,13]. Another type of biocomposite that has attracted much research interest in recent years is the hydroxyapatite(HAp)–collagen composite, which is the focus of this paper [14,15]. 

HAp is the main mineral component of human bones and teeth, and it is a calcium phosphate compound with a chemical formula of Ca_10_(PO_4_)_6_(OH)_2_ [11,16,17,18]. HAp and collagen are a pair of indispensable constituents present in the extracellular matrix within osseous tissue, fulfilling divergent functions in the architecture and operation of skeletal structures. Comprehending the biomechanical characteristics of bone and its capacity to endure both tensile and compressive forces necessitates a thorough understanding of the interrelation between HAp and collagen. Discrepancies in the production or breakdown of Hap and collagen have the potential to result in bone ailments and influence the strength and integrity of bone. Ensuring optimal bone health requires maintaining a nutritious diet that includes adequate quantities of calcium and vitamin D.

Type 1 collagen is the most abundant collagen in the human body, accounting for about 90% of the collagen in our bones [19]. It is a fibrous protein that provides strength and support to our bones, skin, tendons, and other connective tissues. Type 1 collagen has been studied for its potential use in biomedical applications, such as wound healing and tissue engineering [20,21]. HAp–collagen composites combine the advantages of both HAp and type 1 collagen, offering a biocompatible and osteoconductive material that can mimic the structure and function of natural bone tissue [14,22,23]. 

Several methods have been developed to synthesize HAp–collagen composites, and research has focused on various aspects of these composites, such as mechanical properties, biological activity, and degradation [24,25]. The synthesis methods for HAp–collagen composites can be classified into four main categories: sol–gel method, electrospinning method, melt-blending method, and co-precipitation method. Each method has its own advantages and disadvantages depending on the intended application and desired properties of the final composite. Therefore, the selection of the synthesis method depends on the specific requirements of the composite and the expected outcomes of the final product.

The sol–gel method involves adding a solution containing calcium and phosphate ions to a collagen solution, resulting in the formation of HAp–collagen composites [26]. This method can produce HAp–collagen composites in various forms, such as thin films, coatings, and hydrogels [27]. The sol–gel method offers better control over the composition and microstructure of the composite, but it may require high temperatures or pressures for the gelation process. The electrospinning is another method that can produce different forms of HAp–collagen composites. It involves simultaneously electrospinning a solution containing collagen and a precursor solution containing calcium and phosphate ions to form HAp–collagen composites in the form of fibers [28]. This method can produce highly aligned and porous structures that are suitable for tissue engineering applications. However, controlling the electrospinning process can be challenging, unlike the melt-blending method. In the melt-blending method, HAp and collagen powders are melt-blended and then cooled to form HAp–collagen composites in the form of pellets or granules. This method is simple, cost-effective, and allows for a high degree of control over the final composition and morphology of the composite. However, it may not be suitable for certain applications, such as tissue engineering, which may require more complex structures, such as scaffolds. The co-precipitation method is a technique that involves mixing calcium and phosphate ions under controlled conditions to form HAp and then adding collagen to the HAp during the precipitation process, resulting in the formation of HAp–collagen composites [29]. This technique can produce HAp–collagen composites in various forms, such as powders, fibers, and scaffolds. The co-precipitation method is simple and can be performed at room temperature, but it may have some drawbacks, such as the lack of control over the final particle size and shape of the composite, which may affect its mechanical and biological properties [30].

In this study, we propose a novel co-precipitation method for synthesizing HAp–collagen composites with potential application as an orthopedic material for bone regeneration. We used HAp nano-powder and type I collagen with varying mass percentages of collagen to HAp. We characterized the composites using several techniques, such as attenuated total reflectance–Fourier-transform infrared spectroscopy (ATR-FTIR), X-ray diffraction analysis (XRD), thermogravimetric analysis (TGA), scanning electron microscopy (SEM) with energy-dispersive X-ray spectroscopy (EDX), and antimicrobial activity assessment. In this paper, we compare our results with those obtained from other synthesis methods and discuss the advantages and limitations of our approach. The antimicrobial activity of the nanocomposites was also evaluated in vitro against several bacterial and fungal strains to demonstrate their medical potential. In addition, in silico analyses were performed to predict the absorption, distribution, metabolism, and excretion properties and the bioavailability of the collagen samples.

## 2. Materials and Methods

### 2.1. Materials

The reagents used in this study were calcium hydroxide Ca(OH)_2_ (99%), calcium nitrate tetrahydrate Ca(NO_3_)_2_, 4H_2_O (99%), collagen type I (collagen from bovine Achilles tendon), and ammonium hydrogen phosphate (NH_4_)_2_HPO_4_ (99%). They were all purchased from Sigma-Aldrich, Saint-Quentin-Fallavier, France, and were used without further purification. 

### 2.2. Attenuated Total Reflectance–Fourier Transform Infrared Spectroscopy (ATR-FTIR)

ATR-FTIR was used to evaluate the formation of HAp in the composites. The spectra were obtained using a FT/IR-4700 spectrometer (JASCO International LTD., Tokyo, Japan). The spectral range was 400–4000 cm^−1^, and each spectrum was an average of 32 scans at a resolution of 4 cm^−1^. A background check was performed before each analysis.

### 2.3. X-ray Diffraction Analysis with Energy-Dispersive X-ray Spectroscopy (XRD-EDX)

XRD-EDX was used to investigate the crystal structure and the elemental composition of the composites. The XRD patterns were recorded using a Panalytical X’Pert Pro instrument (Malvern Panalytical GmbH, Kassel, Germany) with a Cu Kα radiation source at 40 kV and 30 mA, and a scan rate of 2 °/min. The EDX spectra were obtained using an attached detector on the same instrument. 

To determine the crystallite size and width/thickness of the sample, two XRD diffraction peaks, (102) and (211), were used. The crystallite size was obtained by using the following formula:D = 0.9λ/βcosθ(1)
where D is the crystallite size (nm), β is the full width at half maximum of the peak, λ is the radiation wavelength (0.154 nm), and θ is the diffraction angle of the associated plane (*hkl*).

### 2.4. Thermogravimetric Analysis (TGA)

The composite powder was subjected to standard thermogravimetric analysis, using DTG60 instruments (Shimadzu, Duisburg, Germany) to determine its thermal stability. The analysis was conducted in a nitrogen (N_2_) atmosphere at a heating rate of 10 °C/min, from room temperature to 900 °C.

### 2.5. Scanning Electron Microscopy (SEM) with Energy-Dispersive X-ray Spectroscopy (EDX)

The SEM images were obtained using a Thermo Scientific^TM^ Quattro ESEM (ThermoFisher Scientific, Paisley, United Kingdom) at an accelerating voltage of 15 kV. The films were sputter-coated with gold before imaging to improve the electron beam conductivity. The EDX spectra were acquired using an attached detector on the same instrument.

### 2.6. Molecular Docking and Structure Preparation for HAp–Collagen Complexes

We obtained the 3D molecular structure of collagen type 1 as a triple-helix region (PDB ID: 7CWK) from the Protein Data Bank, which was determined using X-ray crystallography with a resolution of 1.54 Å. We focused on chain A of collagen, which has a molecular weight of 2.4 kDa, and prepared it for molecular modeling, using the PyMol interface. To generate the structure of hydroxyapatite (HAp), we used the HAp unit cell car file, which was available from the INTERFACE-MD (version 1.5) software developed by the Heinz group at the University of Colorado at Boulder. The HAp unit cell consists of six PO_4_^3−^ groups, two OH^−^ groups, and ten Ca^2+^ ions. We designed HAp slabs that contained various numbers of unit cells, using the TopoTools v.1.8 VMD plugin. These slabs simulated the Col 1, Col 2, and Col 3 complexes, which correspond to 5, 10, and 30% of the collagen content, respectively. The molecular weights of the slabs were 48, 24, and 8 kDa. 

To perform molecular docking, we used the AutoDock and AutoDock Vina (ADVina) programs with default settings and employed genetic and gradient optimization algorithms. We identified the collagen–HAp binding site at the HAp center for each protein-HAp complex. To prepare the structures for molecular docking, we assigned Gasteiger partial charges and defined rotatable bonds according to standard protocols published elsewhere [31,32].

### 2.7. In Silico ADME Screening 

To estimate the individual ADME (absorption, distribution, metabolism, and excretion) behaviors of the compounds, the SwissADME software (http://www.swissadme.ch; accessed on 28 September 2023) [33] provided by the Swiss Institute of Bioinformatics [34] was accessed via a web server. The submission page of SwissADME, available through Google, was used for this purpose. Within the software, there is an input zone featuring a molecular sketcher based on Chem Axons Marvin JS [35]. The structure was then transferred to the right-hand side of the submission page, serving as the input for computation. The list is structured in a way that each line represents an input molecule, with multiple inputs being defined using the simplified molecular input line entry system (SMILES). The results are presented for each molecule in the form of tables, graphs, and an Excel spreadsheet. The SwissADME output file consists of a dedicated panel for each molecule, ensuring a clear output and facilitating export of all molecule-related information [36].

The following parameters were calculated to determine the physicochemical properties of the synthetized compounds, using Open Babel version 2.3.0 [37,38]: molecular formula, molecular weight, number of heavy atoms, number of aromatic heavy atoms, fraction, number of rotatable bonds, number of H-bond acceptors, number of H-bond donors, molar refractivity, and topological polar surface area (TPSA). The TPSA method utilizes the summation of tabulated surface contributions of polar fragments, providing a new approach for determining the molecular polar surface area.

### 2.8. Composite Synthesis the Dissolution/Precipitation Method

#### 2.8.1. HAp Synthesis 

The synthesis method utilized was the dissolution/precipitation of HAp (Figure 1). This involved dissolving 1 g of HAp powder in 50 mL of distilled water (prepared as described by [39]), followed by acidification with HNO_3_ to break down the apatitic network (dissolution). The pH of the solution was then raised to 9 by adding NH_4_OH, which resulted in the re-precipitation of nanometric-sized crystals of HAp.

#### 2.8.2. Composite Synthesis

Figure 2 illustrates the synthesis of the HAp–collagen composite via the dissolution/precipitation method. Solution A was prepared by dissolving HAp in 50 mL of distilled water and 0.5 mL of HNO_3_. On the other hand, Solution B consisted of type I collagen dissolved in 100 mL of 0.5 M acetic acid (CH_3_COOH), which was stirred for 12 h at 10 °C until complete dissolution was achieved. 

Solution A was then added dropwise to Solution B, while the pH was adjusted to 9, using NH_4_OH.

The resulting mixture was stirred for 2 h at 25–30 °C. To obtain the composite solution, the mixture was filtered, washed with distilled water, and then dried in an oven at 30 °C for 6 h. This resulted in the desired composite in the form of a white powder.

Different percentages of HAp and collagen were taken from Solution A and Solution B, as shown in Table 1.

### 2.9. Study of the Antimicrobial Activity

#### 2.9.1. Antibacterial Activity

The antibacterial activity of the composites was evaluated in vitro against four bacterial strains: two strains of Gram-negative bacteria (*Escherichia coli* ATCC 10536 and *Pseudomonas aeruginosa* ATCC 49189) and two strains of Gram-positive bacteria (*Listeria monocytogenes* ATCC 12117 and *Staphylococcus aureus* ATCC 6538). These strains were obtained from the Laboratory of Bioresources, Biotechnologies, Ethnopharmacology, and Health (LBBEH), Faculty of Sciences, Oujda. The bacteria were maintained in Muller-Hinton broth (MHB; BIOKAR, France) [40]. A liquid bacterial suspension of 1 mL was added to 9 mL of MHB and incubated at 37 °C for 24 h until the microbial suspension reached the exponential growth phase. Bacterial cultures were adjusted to 0.5 McFarland’s standard to achieve a concentration of 1.5 × 10^8^ CFU/mL. The well method was used to evaluate the antibacterial activity. This involved punching the Muller–Hinton agar (MHA) inoculated with the test bacteria (100 µL) to create wells, which were then filled with 60 µL of extract. Negative controls were performed using dimethyl sulfoxide (DMSO), and positive controls using gentamicin. The cultures were incubated for 18 h at 37 °C after a 30-min pre-diffusion at room temperature [41]. The zone of inhibition around the well was measured in mm, using a sliding caliper, after incubation.

#### 2.9.2. Antifungal Activity

The antifungal activity of the composites was evaluated against three strains from the LBBEH. The strains included *Aspergilus niger*, *Penicillium digitatum*, and *Rhodotorula glutinis*. Petri dishes were prepared by adding 25 mL of sterile Potato Dextrose Agar (PDA) medium for the microorganisms and allowing it to solidify. Then, 100 μL of each microorganism (containing 1 × 10^5^ spores) was plated onto the agar dishes and left to dry for 15 min. Wells with a diameter of 6 mm were created on the agar dishes and were made using a sterile Pasteur pipette [42]. The wells were filled with 60 μL of the extract, while Cycloheximide was used as a positive control, and DMSO as a negative control. The dishes were then incubated at 25 °C for 18–48 h. After incubation, the zone of inhibition around each well was measured in mm, using a sliding caliper. 

## 3. Results and Discussion

### 3.1. Chemical Composition of Composites

The ATR-FTIR spectra show vibrational bands corresponding to OH, NH, CH, and PO_4_^3−^ groups in the synthesized composites (Figure 3). The band associated with the C-C and C-O bond at approximately 1093.71 cm^−1^ overlaps with the PO_4_^3−^ group. Additionally, contributions from COO^−^, C=O, amine (I), and amine (I) groups of collagen and from the CO_3_^2−^ of HAp were observed in the range between 1750 and 1500 cm^−1^. The intensity of the bands located between 2800 and 3000 cm^−1^ attributed to C-H bonds increases with the percentage of collagen in the composite. Notably, changes after mineralization occur mainly in the range from 500 to 1700 cm^−1^. The absorption bands of amides I, II, and III were significantly weakened, and the bands of II and III almost disappeared, indicating the formation of a bond between Ca^2+^ and R-COO^−^ of the collagen molecules [30]. In summary, chemical bonds were formed through the reaction between Ca^2+^ on HAp and -COO^−^ on collagen.

### 3.2. X-ray Diffraction Analysis of Composites

Figure 4 shows the XRD spectra of HAp–collagen composites (Col 1, Col 2, and Col 3) and HAp (Col 0) synthesized using the co-precipitation method. The XRD spectrum of sample Col 0 indicates that it is poorly crystallized HAp [43]. The characteristic peaks of HAp (Col 0), including the most intense peak at 2θ = 31.78°, the peak at 2θ = 25.70°, and other peaks between 2θ = 40° and 2θ = 55°, are detected. The XRD spectrum of type I collagen (Col 4) exhibited a broad peak in the range of 15°~30°, indicating its amorphous structure.

In the composites (Col 1, Col 2, and Col 3), the apatitic structure was preserved despite the increase in the fraction of collagen. However, with an increase in the collagen percentage, the peaks at about 26° and 32° weakened, and the broad peaks at about 15° and 30° gradually emerged, suggesting that the components of the composites were adjusted by the preparatory parameters.

As shown in the XRD diagram, the full width at half maximum of the indexed lines (002) and (310) was used to determine the average crystallite size, using the Scherrer formula. The dimensions of the crystallites in directions perpendicular to the planes (002) or (310) are reported in Table 2, indicating that the particle size of HAp is approximately 22.62 nm, and it is 14.81 nm for HAp–collagen. These results confirm the nanometric nature of the HAp and HAp–collagen particles.

### 3.3. Thermal Stability of Composites 

The significance of temperature in the crystalline transformations and matter transfers during thermal treatment of HAp (Col 0), collagen (Col 4), and composites (Col 1, Col 2, and Col 3) can be demonstrated by comparing their respective behaviors. To investigate this, TGA tests were conducted over a temperature range of room temperature to 850 °C. Figure 5 depicts the thermogravimetric analysis curves for commercial type-I collagen, synthetic HAp, and the synthesized composites.

Table 3 presents the remaining and residual masses in percentages for commercial type I collagen, synthesized HAp, and the three synthesized composites.

The mass losses presented in Figure 5 correspond to the removal of water molecules adsorbed on the HAp surface in the first stage, and the release of amino acid chains from collagen in the second stage. At this temperature, collagen can be assumed to detach from the apatite matrix. Among the three composites, there is a noticeable difference in the amount of mass degraded. As shown in Table 3, the Col 3 composite (20.6946%) undergoes more degradation than the Col 1 (9.0981%) and Col 2 composites (17.8542%).

The Col 1 and Col 2 composites display the smallest amount of mass loss, which can be explained by the respective amounts of HAp in each composite. Thus, the greater the amount of collagen in the material, the greater its mass loss. Furthermore, an increase in the fraction of HAp in the preparatory recipe led to an increase in the fraction of the residue in the products.

### 3.4. SEM-Based Morphological Study

The scanning electron microscopy was used to study the size, dispersion, and shape of apatite particles on the surface of various synthetic products. Figure 5 displays the presence of nanoscale particles in composites as agglomerates, with nonuniform particle shapes (Figure 6). To assess the homogeneity of the HAp–collagen distribution, SEM-EDX mapping was conducted to map the elemental distribution on the surface of sample Col 3 (Figure 7). This analysis revealed many interesting aspects regarding the distribution of elements.

The signal intensity of each ion is reflected by the intensity of each image. As shown in Figure 7, Ca, P, N, O, and C particles are represented by different colors on the distribution maps. The distribution of composite particles (Col 3) was found to be heterogeneous. The results indicate an increase in the percentage of carbon (C) and nitrogen (N) with the percentage increase in collagen. The Ca/P ratio between 1.65 and 1.53 confirmed the formation of non-stoichiometric apatites in all samples.

### 3.5. Microbial Activity

The antimicrobial activity of HAp–collagen composites was evaluated through the sensitivity tests (Table 4). Our findings indicated that the mass percentage ratios of collagen and HAp significantly influenced the inhibition rate, as shown in Figure 8 and Table 4. Specifically, the HAp–collagen composite effectively inhibited the growth of *P. digitatum* and *R. glutinis*, with an inhibition diameter of approximately 14.3 and 11.0 mm, respectively. Additionally, the composite displayed strong inhibition of *A. niger*, with an inhibition diameter of 19.2 mm. Notably, growth inhibition increased with higher collagen percentages. However, we did not observe any antibacterial activity with the HAp–collagen composite (Figure 8).

### 3.6. Molecular Modeling

To investigate the binding between collagen and HAp and its correlation to microbial inhibition, we used different molecular sizes of HAp with collagen percentages of 5, 10, and 30% (Figure 9). Recent in silico studies have shed light on the molecular mechanisms underlying this interaction, revealing key binding sites and interactions between collagen and HAp at the atomic level [44,45,46]. Our computational analyses showed that Col 1 had the lowest AutoDock and ADVina binding energies compared to the other complexes (Table 5). This observation may be explained by the high stability of the Col 1 composite, as confirmed by TGA experiments, where it showed the lowest percentage of degradation (9.09%) due to the bond formation between Ca^2+^ of HAp and -COO^−^ of collagen. Additionally, Col 1 exhibited the highest microbial inhibition, with an MI_total_ of 43.01%. It was the only composite that demonstrated a strong antibacterial effect against both *E. coli* and *P. aeruginosa*. 

Furthermore, we calculated the solvent-accessible surface area (SASA) and radius of gyration (R_g_) for collagen binding poses obtained from the Col 1, Col 2, and Col 3 complexes to correlate them with the MI and affinity parameters. Our findings revealed that Col 1 exhibited the highest SASA, which can be attributed to its lowest Gibbs free energy of binding, suggesting a larger intermolecular interface than that of the Col 2 and Col 3 poses (Table 6). Interestingly, the Rg values for Col 1 and Col 2 were similar, while SASA remained different (Table 6 and Figure 10). This discrepancy may be due to changes in the surface conformation or protrusions of the biomolecule without altering its overall size or shape.

### 3.7. In Silico ADME Predictions

The pharmacological or therapeutic effect of a drug depends on how its physicochemical properties influence the biomolecule that it binds to. In silico approaches are widely used in drug discovery to assess the ADME properties of compounds at the early stages of discovery to identify potential lead molecules. Different physicochemical parameters of drug candidates affect their pharmacokinetic behavior.

Therefore, calculating and measuring these parameters can help prioritize compounds for screening as efficient drug candidates and avoid premature decisions in drug discovery [22]. A molecule that is likely to be developed as an orally active drug should follow the Lipinski rule of five [23], which states the following four criteria: partition coefficient (Clog P) ≤ 5, molecular weight (MW) ≤ 500, number of hydrogen bond acceptors (HBA) ≤ 10, and number of hydrogen bond donors (HBD) ≤ 5. Violating more than one of these rules would result in poor bioavailability upon oral administration. According to Veber et al. [24], the number of rotatable bonds should be ≤10, which is an indicator of good bioavailability. In this study, we calculated several parameters to predict the drug-likeness properties of collagen to screen potential candidate drugs. The molecule was subjected to in silico physicochemical studies such as number of rotatable bonds (nROTB), HBA, HBD, lipophilicity (iLogP), and topological polar surface area (TPSA), which were calculated to understand the drug’s transport properties. The in-silico percentage absorption was calculated using the formula reported by Hou et al.: (%ABS = 109 − (0.345 × TPSA)) [25]. The results are shown in Table 7.

The collagen samples complied with the Lipinski rule of five, as their molecular weight (MW) ranged from 310 to 410, well below the threshold of 500. Additionally, these collagen samples demonstrated favorable drug-likeness properties, including hydrogen bond acceptors (HBA) ranging from 3 to 7 (≤10), hydrogen bond donors (HBD) in the range of 0–1 (≤5), and iLogP (lipophilicity) ranging between 2.69 and 3.61 (≤5). These characteristics suggest that, upon administration, these collagen samples possess properties suitable for drug-like molecules. Furthermore, all collagen samples complied with the Veber rule, with a range of rotatable bonds (nROTB) falling between 5 and 7 (<10), indicating good bioavailability. These findings highlight the potential of collagen samples for absorption and their suitability for therapeutic applications.

## 4. Conclusions

In this study, we designed an innovative co-precipitation method to synthesize composites based on HAp and collagen by varying the ratios of the two components. We carried out a comprehensive characterization of the composites, assessing their crystal structure, thermal stability, and antimicrobial activity. In addition, in silico methods were employed to anticipate the absorption, distribution, metabolism, and excretion properties of collagen samples. This study therefore presents an in-depth approach to the design, synthesis, and characterization of biocompatible composites combining HAp and collagen, paving new avenues for biomedical applications, particularly in the fields of bone regeneration and orthopedic implantology. Future research directions will include assessing the biocompatibility and bioactivity of nanocomposites in vivo, while exploring other potential applications and functionalities of these innovative materials.

## Figures and Tables

**Figure 1 polymers-16-00085-f001:**
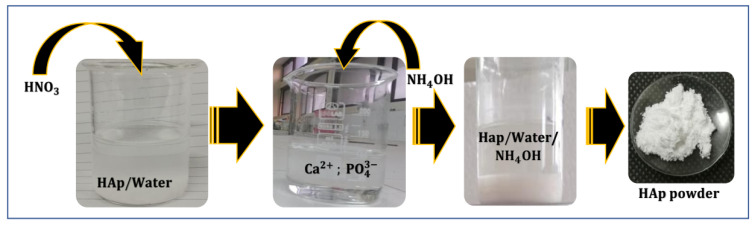
Schematic representation of HAp synthesis via the dissolution/precipitation method.

**Figure 2 polymers-16-00085-f002:**
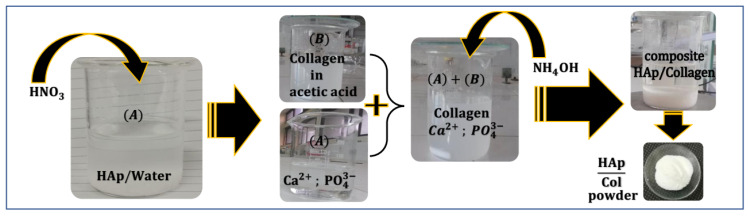
Schematic representation of HAp–collagen composite synthesis via the dissolution/precipitation method.

**Figure 3 polymers-16-00085-f003:**
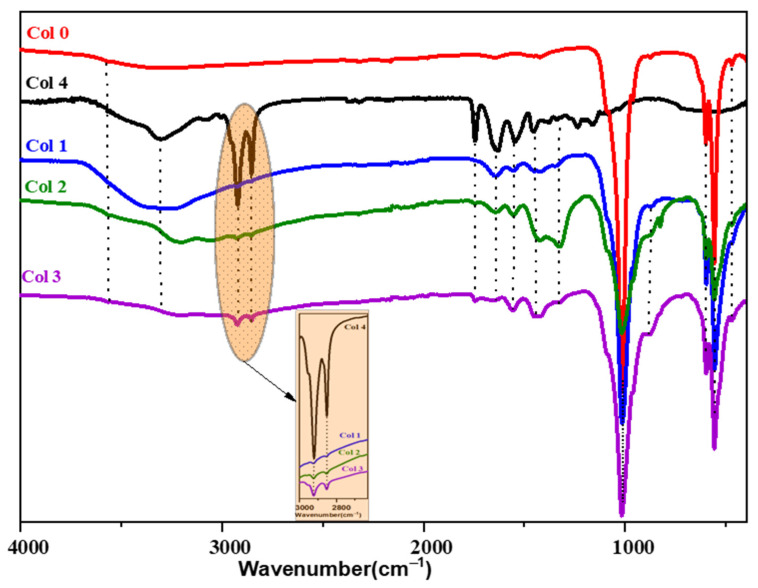
ATR-FTIR spectra of hydroxyapatite (Col 0), collagen type I (Col 4), and hydroxyapatite/collagen/composites (Col 1, Col 2, and Col 3).

**Figure 4 polymers-16-00085-f004:**
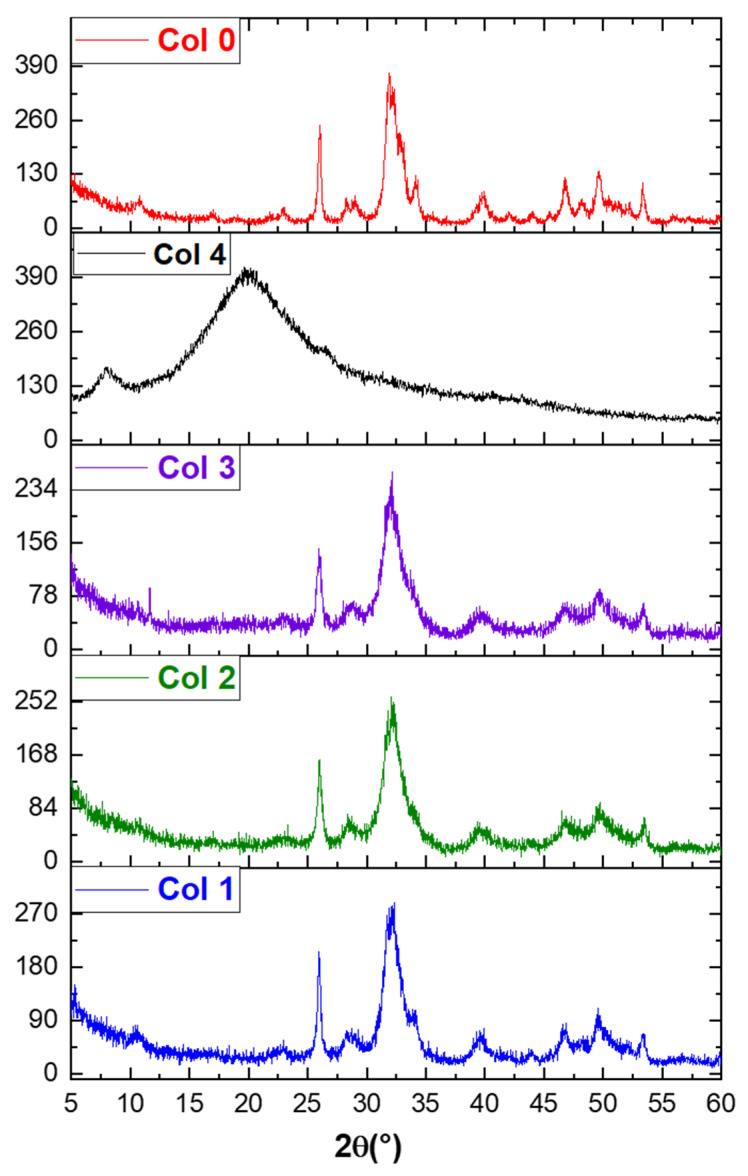
XRD spectra of hydroxyapatite (Col 0), collagen type I (Col 4), and hydroxyapatite/collagen/composites (Col 1, Col 2, and Col 3).

**Figure 5 polymers-16-00085-f005:**
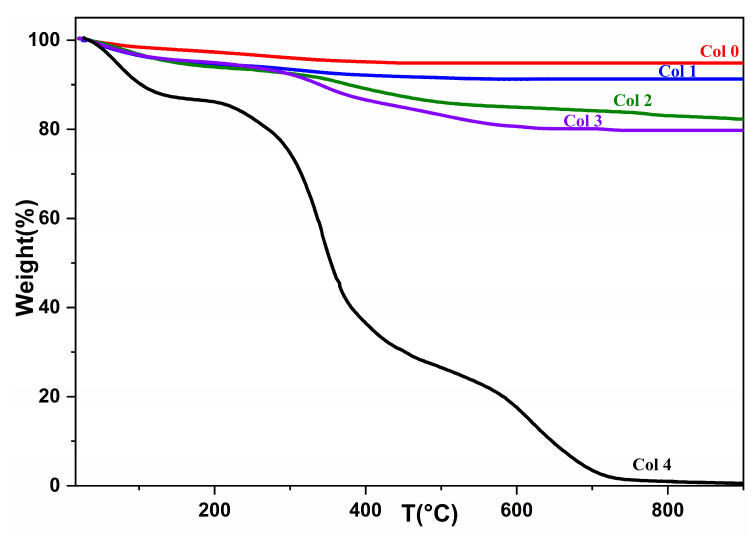
TGA spectrum of hydroxyapatite (Col 0), collagen (Col 4), and composites (Col 1, Col 2, and Col 3).

**Figure 6 polymers-16-00085-f006:**
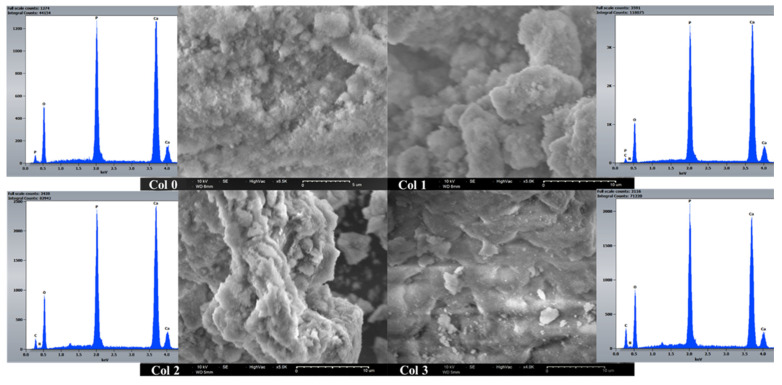
SEM-EDX analysis of composites (Col 0, Col 1, Col 2, and Col 3).

**Figure 7 polymers-16-00085-f007:**
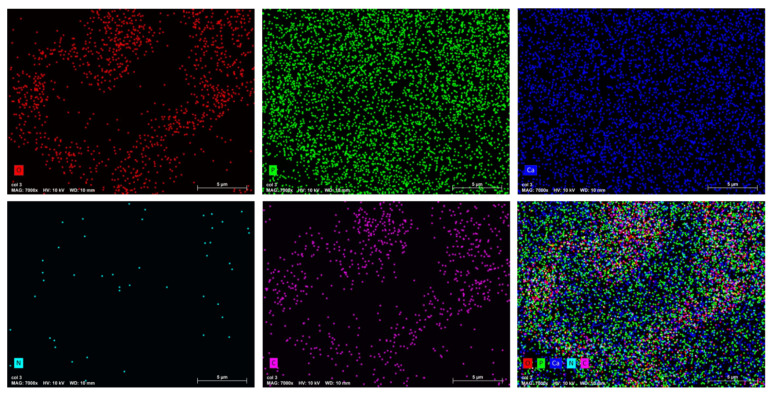
EDX mapping of elements O, P, Ca, N, and C in the synthesized Col 3 composite.

**Figure 8 polymers-16-00085-f008:**
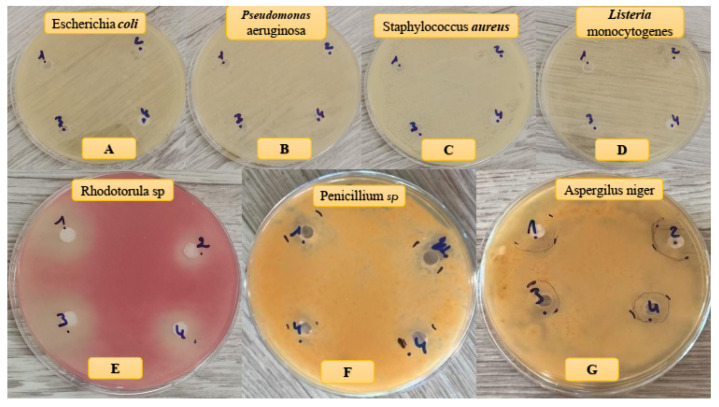
Antibacterial (**A**–**D**) and antifungal (**E**–**G**) effects of samples on different bacteria and fungi strains.

**Figure 9 polymers-16-00085-f009:**
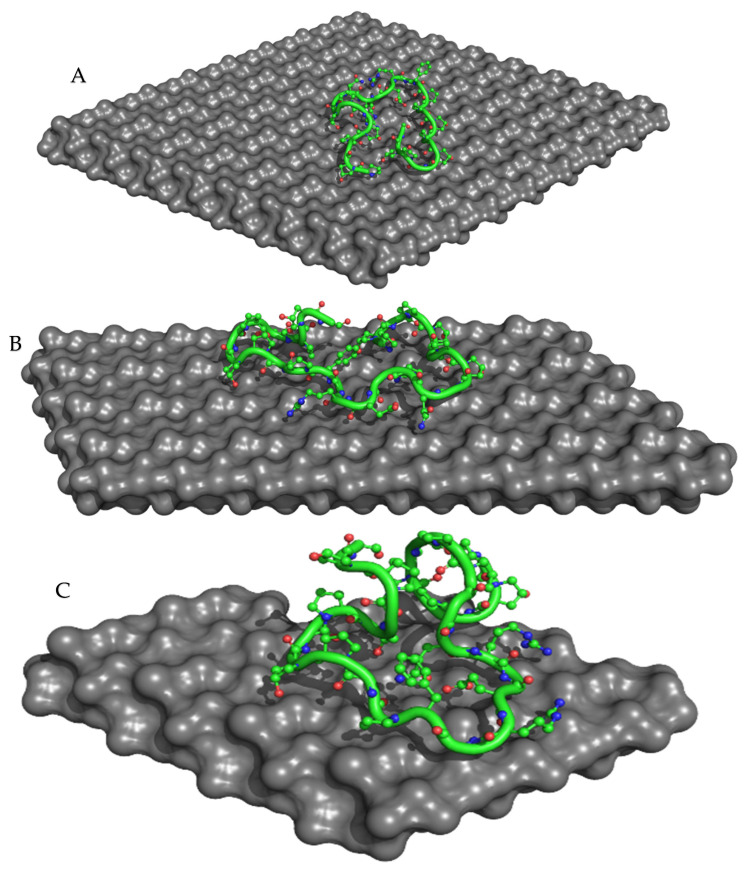
ADVina-predicted interaction between collagen type 1 and the HAp molecule for the Col 1 (**A**), Col 2 (**B**), and Col 3 (**C**) complexes. The protein molecule is presented in both cartoon and ball-and-stick representations, with colors corresponding to its atomic composition. For clarity purposes, all hydrogen atoms are omitted from the visualization.

**Figure 10 polymers-16-00085-f010:**
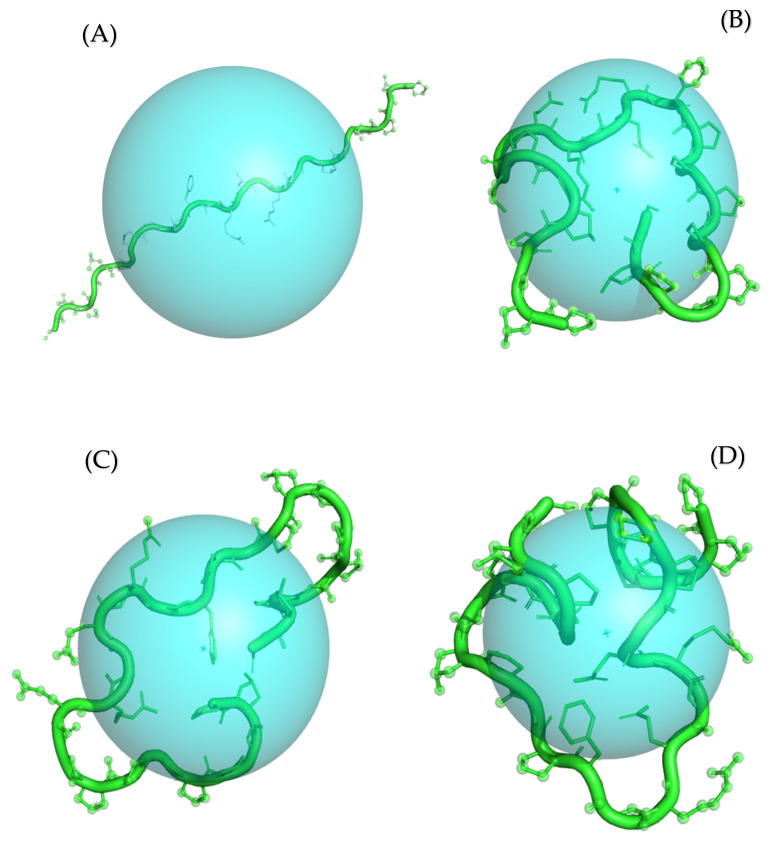
Calculated radius of gyration (Rg) for Ref (**A**), Col 1 (**B**), Col 2 (**C**), and Col 3 (**D**) conformations of collagen type 1. The Rg parameter is visualized as a sphere with the center corresponding to the center of mass of the molecule. The protein molecule is presented in both cartoon and ball-and-stick representations, colored in green. For clarity purposes, all hydrogen atoms are omitted from the visualization.

**Table 1 polymers-16-00085-t001:** Powder synthesis of various composites.

	Collagen Type I (%)	Hydroxyapatite (%)
Col 0	0	100
Col 1	5	95
Col 2	10	90
Col 3	30	70
Col 4	100	0

**Table 2 polymers-16-00085-t002:** Crystallite size and lattice parameters of HAp and HAp–collagen composites.

Compound	Plane (*hkl*)	D_mean_ (nm)
Col 0	(0 0 2)	22.62 ± 2.10
	(3 1 0)	
Col 2	(0 0 2)	21.49 ± 2.30
	(3 1 0)	
Col 3	(0 0 2)	17.37 ± 3.00
	(3 1 0)	
Col 4	(0 0 2)	14.81 ± 2.50
	(3 1 0)	
	(3 1 0)	

**Table 3 polymers-16-00085-t003:** Mass loss of composites, hydroxyapatite, and collagen between room temperature (T_1_ = 25 °C) and 850 °C.

	Residue (%)	Degradation (%)
Col 0	92.0717	7.9283
Col 1	90.9019	9.0981
Col 2	82.1458	17.8542
Col 3	79.3054	20.6946
Col 4	0.0889	99.9111

**Table 4 polymers-16-00085-t004:** Inhibition diameter (mm) of composites tested on three fungal and four bacterial strains.

	Col 0	Col 1	Col 2	Col 3	Cycloheximide	DMSO
*Penicillium digitatum*	17.1 ± 0.4	13.5 ± 0.2	11.5 ± 0.2	14.3 ± 0.2	29.1 ± 0.3	00 ± 00
*Aspergilus niger*	12.1 ± 0.1	15.1 ± 0.2	18.03 ± 0.3	19.2 ± 0.1	22.3 ± 0.02	00 ± 00
*Rhodotorula glutinis*	13 ± 0.3	11.3 ± 0.1	10.5 ± 0.2	11 ± 0.1	28.1 ± 0.1	00 ± 00
*Escherichia coli*	7.8 ± 0.1	8.1 ± 0.3	00 ± 00	00 ± 00	31 ± 0.1	00 ± 00
*Pseudomonas aeruginosa*	00 ± 00	7.2 ± 0.1	00 ± 00	00 ± 00	21 ± 0.2	00 ± 00
*Staphylococcus aureus*	8 ± 0.2	00 ± 00	00 ± 00	00 ± 00	30.7 ± 0.3	00 ± 00
*Listeria monocytogenes*	00 ± 00	00 ± 00	00 ± 00	00 ± 00	30.3 ± 0.2	00 ± 00

**Table 5 polymers-16-00085-t005:** Gibbs free energy of binding (ΔG_bind_) determined by the genetic (AutoDock) and gradient optimization (ADVina) algorithms, inhibition constants (Ki), and total microbial inhibition (MI_tot_) for collagen type 1 bound to HAp.

Complex	Binding Energy (Kcal/mol)		Ki (µM)	MI_tot_ (%)
	${\Delta G}_{bind}^{AD}$	${\Delta G}_{bind}^{AV}$		
Col 1	−45.96	−8.2	0.98	43.01
Col 2	−32.74	−7.9	1.63	31.62
Col 3	−15.69	−5.7	66.68	34.95

Total MI was determined as a sum of two bacterial (*E. coli* and *P. aeruginosa*) and three fungal MIs in %.

**Table 6 polymers-16-00085-t006:** Solvent-accessible surface area (SASA) in Å^2^ and radius of gyration (Rg) in Å calculated for reference (Ref) molecule of collagen type 1 and its binding poses obtained from the Col 1, Col 2, and Col 3 complexes.

Pose	SASA_avg_	SASA_sum_	Rg
Ref	1.03	367.01	21.92
Col 1	0.82	288.85	10.86
Col 2	0.76	267.77	10.97
Col 3	0.62	218.41	8.41

**Table 7 polymers-16-00085-t007:** In silico ADME predictions of the collagen samples.

Physicochemical Properties		Lipophilicity		Water Solubility		
Formula	C_57_H_9_1N_19_O_16_	Log Po/w (iLOGP)	2.94	Log S (ESOL)	−1.08	
Molecular weight	1298.45 g/mol	Log Po/w (XLOGP3)	−5.13	Solubility	1.09 × 10^2^ mg/mL; 8.38 × 10^−2^ mol/l	
No. of heavy atoms	92	Log Po/w (WLOGP)	−9.02	Class	Very soluble	
No. of arom. heavy atoms	6	Log Po/w (MLOGP)	−7.04	Log S (Ali)	−5.93	
Fraction Csp3	0.61	Log Po/w (SILICOS-IT)	0.74	Solubility	1.53 × 10^−3^ mg/mL; 1.18 × 10^−6^ mol/l	
No. of rotatable bonds	55	Consensus Log Po/w	−3.50	Class	Moderately soluble	
N H-bond acceptors	22			Log S (SILICOS-IT)	−8.20	
No. of H-bond donors	18			Solubility	8.17 × 10^−6^ mg/mL; 6.30 × 10^−9^ mol/l	
Molar Refractivity	339.61			Class	Poorly soluble	
TPSA	557.12 Å^2^					
**Pharmacokinetics**		**Drug-Likeness**			**Medicinal Chemistry**	
GI absorption	Low	Lipinski	No; 3 violations: MW > 500, NorO > 10, NHorOH > 5		PAINS	0 alert
BBB permeant	No	Ghose	No; 4 violations: MW > 480, WLOGP < −0.4, MR > 130, #atoms > 70		Brenk	4 alerts: beta_keto_anhydride, imine_1, imine_2, more_than_2_esters
P-gp substrate	Yes	Veber	No; 2 violations: Rotors > 10, TPSA > 140		Leadlikeness	No; 2 violations: MW > 350, Rotors > 7
CYP1A2 inhibitor	No	Egan	No; 1 violation: TPSA > 131.6		Synthetic accessibility	10.00
CYP2C19 inhibitor	No	Muegge	No; 6 violations: MW > 600, XLOGP3 < −2, TPSA > 150, Rotors > 15, H-acc > 10, H-don > 5			
CYP2C9 inhibitor	No	Bioavailability Score	0.17			
CYP2D6 inhibitor	No					
CYP3A4 inhibitor	No					
Log Kp (skin permeation)	−17.86 cm/s					

## Data Availability

The data presented in this study are available upon request from the corresponding author.

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
