# Peer review of "Cheminformatics-Based Design and Synthesis of Hydroxyapatite/Collagen Nanocomposites for Biomedical Applications"

_polymers, 2023, doi:10.3390/polym16010085_

Round 1

Reviewer 1 Report

Comments and Suggestions for Authors

The Authors' did some interesting work using HAp and Collagen. Here is my comments. 

1. The abstract should be more focused. Authors should have mentioned their important findings (including data) in the abstract. 

2. The introduction section must be improved. 

3. The authors mixed Hap Solution with Collagen solution where the HAp solution contained HNO3. Such strong acid could degenerate the collagen structure. The authors should check this fact. 

4. The authors claim that "The intensity of the bands located between 2800 and 3000 cm-1 attributed to C-H bonds increases with the percentage of collagen in the composite." - The FTIR figure does not show any significant change in the peak structure. 

5. The intensity at which the XRD was conducted should be mentioned. Y-axis should be added. Also, this graph hardly gives any idea of the crystallinity. The authors' should measure the crystallinity index to give more specific idea on the crystallinity.

6. TGA shows very expected data. The degradation is supposed to increase with increase in collagen percentage. 

7. SEM images does not give proper idea of the morphology. 

8. In the conclusion, the author mentioned that, HAp- and collagen-based composites could open new possibilities for biomedical applications, particularly in bone regeneration and orthopedic implantology. However, they did not do any compressive strength or mechanical strength tests. It would be great if they could explain the reason behind such claim. Also, what is the point of showing antimicrobial/bacterial property in the manuscript since there is no changes due to use of the composite.  

Comments on the Quality of English Language

Not applicable. 

Author Response

Submission of the revised manuscript

Journal: polymers

Manuscript Number: polymers-2767470

Title of paper: Cheminformatics-Based Design and Synthesis of Hydroxyapatite/Collagen Nanocomposites for Biomedical Applications.

Authors: M. Aaddouz, K. Azzaoui, R. Sabbahi, M H. Youssoufi, M. Idrissi Yahyaoui, A. Asehraou, M. El Miz, B. Hammouti, S. Shityakov, M. Siaj, E. Mejdoubi

Response:

Thank you very much for your letter and for the reviewers' comments on our manuscript. These comments are all valuable and very helpful in revising and improving our manuscript, as well as the importance of our research direction. We have carefully studied the comments and made corrections which we hope will be approved. Below is a point-by-point response to each of the comments made. All changes are marked in yellow on the revised copy.

Reviewer #1:

The Authors did some interesting work using HAp and Collagen. Here is my comments. 

  1. The abstract should be more focused. Authors should have mentioned their important findings (including data) in the abstract. 

Response: We extend our appreciation to the reviewer for his constructive feedback. Your suggestion has been considered and we have added relevant contents in the abstract.

  1. The introduction section must be improved. 

Response: Thank you for your valuable comment. We have added relevant contents to the introduction.

  1. The authors mixed HAp Solution with Collagen solution where the HAp solution contained HNO3. Such strong acid could degenerate the collagen structure. The authors should check this fact. 

Response: Thank you for your comment. We used HNO3 with a pH of around 3 to prevent collagen type 1 degeneration [1].

  1. Ratanavaraporn, J., et al., Effects of acid type on physical and biological properties of collagen scaffolds. Journal of Biomaterials Science, Polymer Edition, 2008. 19(7): p. 945-952.
  2. The authors claim that "The intensity of the bands located between 2800 and 3000 cm-1 attributed to C-H bonds increases with the percentage of collagen in the composite." - The FTIR figure does not show any significant change in the peak structure. 

Response: Thank you for your comment. The change in intensity is illustrated by the addition of a small spectrum which clearly shows the extent of the increase mentioned (Fig. 3).

  1. The intensity at which the XRD was conducted should be mentioned. Y-axis should be added. Also, this graph hardly gives any idea of the crystallinity. The authors should measure the crystallinity index to give more specific idea on the crystallinity.

Response:  We extend our appreciation to the reviewer for the constructive feedback. The Y-axis Has been added to the XRD graph. We have also calculated the average crystallite size and we have included the results in the paper. We hope that these changes address your concerns and enhance the quality of our work.

  1. TGA shows very expected data. The degradation is supposed to increase with increase in collagen percentage. 

Response: Thank you for your constructive comment. Figure 5 clearly shows that degradation increases with the percentage of collagen, from Col 0 (HAp) to Col4 (100% collagen).

  1. SEM images does not give proper idea of the morphology. 

Response: Thank you for your comment. SEM images provide an idea of the morphology, showing the presence of nanoparticles in aggregate form. To confirm this, we need to move on to TEM (transmission electron microscopy).

  1. In the conclusion, the author mentioned that HAp- and collagen-based composites could open new possibilities for biomedical applications, particularly in bone regeneration and orthopedic implantology. However, they did not do any compressive strength or mechanical strength tests. It would be great if they could explain the reason behind such claim. Also, what is the point of showing antimicrobial/bacterial property in the manuscript since there is no changes due to use of the composite.  

Response: Thank you for your valuable comment. We agree that compressive strength and mechanical strength tests are important for assessing the performance and durability of HAp- and collagen-based composites for bone regeneration and orthopedic implantology. However, we did not carry out these tests as we focused on the morphology, biocompatibility and antibacterial properties of the composites, which are also essential for their potential applications. We will be working on the prospect of biomedical applications, notably in bone regeneration and orthopedic implantology.

Reviewer 2 Report

Comments and Suggestions for Authors

The authors develop a new co-precipitation method to prepare composites of HAp and collagen with different ratios of the two entities. The various biocomposites obtained are characterized (physicochemistry, mechanics, crystalline structure, thermal stability, antimicrobial activity). An in silico study is also carried out to predict the absorption, distribution, metabolism and excretion properties of the collagen samples. Is it not possible to carry out an in silico (AI) study further upstream, and then design the best biocomposite.
 The study is well described in terms of methodology, results.
The authors mention the prospect of in vivo studies with other possible applications; it would be good to be a little more specific.

Other comments;
- discussions are very limited. It is necessary for the authors to discuss their results in comparison with what already exists.
- lines 49-52; how HAp-Collagen biocomposites are better than chitosan-based composites.
- lines 107-108; why not design biocomposites upstream using AI, then synthesize and test them.
- lines 334-335; references should be cited by number.

Author Response

Submission of the revised manuscript

Journal: polymers

Manuscript Number: polymers-2767470

Title of paper: Cheminformatics-Based Design and Synthesis of Hydroxyapatite/Collagen Nanocomposites for Biomedical Applications.

Authors: M. Aaddouz, K. Azzaoui, R. Sabbahi, M H. Youssoufi, M. Idrissi Yahyaoui, A. Asehraou, M. El Miz, B. Hammouti, S. Shityakov, M. Siaj, E. Mejdoubi

Response:

Thank you very much for your letter and for the reviewers' comments on our manuscript. These comments are all valuable and very helpful in revising and improving our manuscript, as well as the importance of our research direction. We have carefully studied the comments and made corrections which we hope will be approved. Below is a point-by-point response to each of the comments made. All changes are marked in yellow on the revised copy.

Reviewer #2:

The authors develop a new co-precipitation method to prepare composites of HAp and collagen with different ratios of the two entities. The various biocomposites obtained are characterized (physicochemistry, mechanics, crystalline structure, thermal stability, antimicrobial activity). An in silico study is also carried out to predict the absorption, distribution, metabolism and excretion properties of the collagen samples. Is it not possible to carry out an in silico (AI) study further upstream, and then design the best biocomposite. The study is well described in terms of methodology, results. The authors mention the prospect of in vivo studies with other possible applications; it would be good to be a little more specific.

Response: Thank you very much for your feedback. We appreciate your suggestion to use an in silico (AI) study to design the best biocomposite for different applications. However, in this study we focused on the experimental characterization of biocomposites and their properties, which are also essential for their potential applications, and an in silico study was also performed to predict the absorption, distribution, metabolism and excretion properties of collagen samples. We will be working on the prospect of biomedical applications, particularly in bone regeneration and orthopedic implantology.

Other comments;

- discussions are very limited. It is necessary for the authors to discuss their results in comparison with what already exists.

Response: Thank you for your constructive comment. Relevant information has been added to the paper.

- lines 49-52; how HAp-Collagen biocomposites are better than chitosan-based composites.

Response: Thank you for your review and valuable comments. Collagen/HAp biocomposites exhibit increased similarity to the bone structure owing to the amalgamation of collagen, a natural component of the bone matrix, with hydroxyapatite. This combination imparts biomimetic characteristics to the biocomposite, rendering it more akin to the natural composition of bone. On the other hand, chitosan is a natural polymer.

- lines 107-108; why not design biocomposites upstream using AI, then synthesize and test them.

Response: Thank you for your comment. Thank you very much for your feedback. We appreciate your suggestion to use an in silico (AI) study to design the best biocomposite for different applications. However, in this study we focused on the experimental characterization of biocomposites and their properties, which are also essential for their potential applications, and an in silico study was also performed to predict the absorption, distribution, metabolism and excretion properties of collagen samples. We plan to conduct an AI-based design study in our future work to explore the possibilities and challenges of this approach.

- lines 334-335; references should be cited by number.

Response: Thank you. The corrections have been made.

Round 2

Reviewer 2 Report

Comments and Suggestions for Authors

As the authors have responded to my questions and comments, I propose publication of the article.